# Semantic Residual Pyramid Network for Image Inpainting

Haiyin Luo [1],* and Yuhui Zheng [2]

1 School of Computer Science, Nanjing University of Information Science & Technology, Nanjing 210044, China
2 Engineering Research Center of Digital Forensics, Ministry of Education, Nanjing University of Information Science & Technology, Nanjing 210044, China; zheng_yuhui@nuist.edu.cn
* Correspondence: 20201220026@nuist.edu.cn

**Abstract:** Existing image inpainting methods based on deep learning have made great progress. These methods either generate contextually semantically consistent images or visually excellent images, ignoring that both semantic and visual effects should be appreciated. In this article, we propose a Semantic Residual Pyramid Network (SRPNet) based on a deep generative model for image inpainting at the image and feature levels. This method encodes a masked image by a residual semantic pyramid encoder and then decodes the encoded features into a inpainted image by a multi-layer decoder. At this stage, a multi-layer attention transfer network is used to gradually fill in the missing regions of the image. To generate semantically consistent and visually superior images, the multi-scale discriminators are added to the network structure. The discriminators are divided into global and local discriminators, where the global discriminator is used to identify the global consistency of the inpainted image, and the local discriminator is used to determine the consistency of the missing regions of the inpainted image. Finally, we conducted experiments on four different datasets. As a result, great performance was achieved for filling both the regular and irregular missing regions.

**Keywords:** image inpainting; residual blocks; multi-scale discriminators; irregular regions inpainting

## 1. Introduction

Image inpainting first originated from an extremely primitive technique in which artists restored a damaged painting in order to match it to the original painting as much as possible [1]. In computer vision, this is realized by filling the missing pixels in the damaged images. Currently, this technique is widely applied in many areas, such as old-photo restoration [1], object removal [2], photo modification [3], and text removal [1].

Existing image inpainting methods can be divided into two categories. The first methods are the traditional methods that were diffusion-based [1,4,5] or patch-based texture synthesis techniques [6,7], which fill the low-level features of the images. Due to the lack of a high-level understanding of the images, such approaches are unable to generate reasonable semantic results. To address this problem, the second methods [8–11] attempt to solve the inpainting problem by a learning-based approach, which predict the pixels in the missing regions by training deep convolution networks. This approach is used to mainly fill the deep features of the images. However, although semantically relevant images can be generated, obtaining visually realistic results is still challenging.

To obtain visually realistic and semantically consistent images, we propose the Semantic Residual Pyramid Network (SRPNet) for filling the missing regions of the images at the image and feature levels. Our work is based on the Pyramid-Context Encoding Network (PEN-NET) [12], which was proposed in 2019. PEN-NET [12] used U-Net [13] as its backbone. However, U-Net has shallow layers and fewer parameters than many current networks, so it is easy to overfit [14] during training. On the other hand, we can obtain more high-level semantic features by arbitrarily increasing the depth of the network. However, increasing the depth of the network is not always applicable due to the non-convergence of

the network caused by the disappearance of the gradient. Therefore, our model introduces the residual blocks [11,15] to address the reduced precision caused by the increase in the depth of the neural network. At the same time, we used the instance normalization [16] to accelerate the convergence of the model. Then, we added the multi-scale discriminators [10] to obtain semantically consistent and visually superior images. This idea improves the refinement of the inpainting results by determining whether the image is consistent with the ground truth. The multi-scale discriminators include a global discriminator and a local discriminator. The global discriminator takes the complete image as input to identify the global consistency of the image, whereas the local discriminator takes the missing regions in the completed image as input to judge the consistency of the missing regions. Experiments demonstrate that this design can obtain richer image details and more realistic images.

Our approach was experimented on four different scene datasets, DTD [17], Facade [18], CELEBA-HQ [19], and Places2 [20]. The experimental results show that this method performs generally well and has good visual effects. Some of the experimental results are shown in Figure 1.

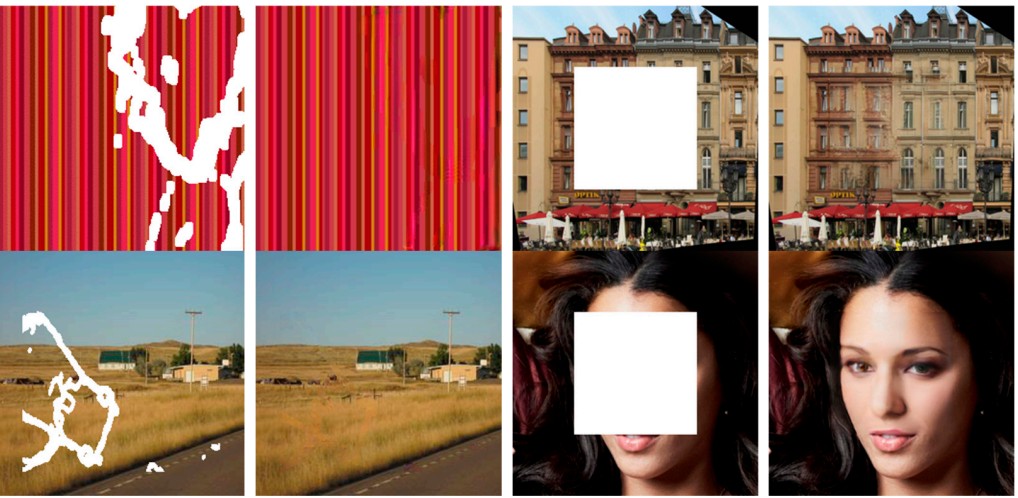

**Figure 1.** Masked images and the corresponding inpainting results generated by the Semantic Residual Pyramid Network (SRPNet).

In brief, the main contributions of this study are as follows:

1. We designed a novel residual pyramid encoder to obtain high-level semantic features by adding the residual blocks to the semantic pyramid encoder;
2. We introduced multi-scale discriminators based on generating adversarial networks to judge whether the semantic features of images at different scales are consistent. Thus, we can obtain richer texture details and semantic features that are more consistent with the ground truth.

The rest of this article is presented as follows. We discuss the related work of this paper in Section 2. Then, we describe the proposed methodological framework in Section 3. The experimental settings and the analysis of experimental results are presented in Section 4. We summarize future directions in Section 5.

## 2. Related Work

### 2.1. Image Inpainting

Existing image inpainting techniques are mainly divided into two categories: traditional approaches based on diffusion or patch synthesis and deep methods that learn semantic features through the convolutional neural network.

Traditional diffusion-based [1,4,5] or patch-based [6,7] texture synthesis methods are the earliest image inpainting methods for filling the missing area at the image layer. They often used patch similarity or differential algorithms to smoothly propagate the image

content from the boundary area to the interior of the missing area. However, traditional image inpainting methods lack a deep understanding of the image, so it is difficult to generate reasonable semantic inpainting results for complex structure images.

Our work focused on deep learning model image inpainting approaches to predict the pixels of the missing regions by training deep convolutional neural networks. The models are divided into Convolutional Neural Networks (CNNs) [21] and Generative Adversarial Networks (GANs) [22] for image inpainting. The main idea was to encode the image for potential features, fill the missing regions in the feature layer, and then decode the acquired features as an image. In recent years, deep learning models have also achieved amazing results. Jain et al. [8] were the first to use CNN for image inpainting. They set the computational task framework as a statistical framework for regression rather than density estimation. However, this approach was limited to greyscale (monochromatic channel) images, and it resulted in significant computational costs. Subsequently, Pathak et al. [9] suggested a deep generative network for predicting the missing content of arbitrary image regions, which was named Context-Encoders. This was used to inpaint a $64 \times 64$ central square region in the $128 \times 128$ image. Lizuka et al. [10] presented a GAN-based image inpainting technique containing a global discriminator and a local discriminator. Although this method can enhance the local consistency of the image, it requires postprocessing of the resulting image from their model to remove the blur. Then, Nazeri et al. [15] proposed a two-stage inpainting model based on adversarial learning. They used the edge structure information of the image to inpaint the image. This approach can realize the inpainting of a highly structured image, but the edge information of the image has a great influence on the subsequent inpainting. In 2018, Yu et al. [23] first proposed to apply the attention transfer model to image inpainting and obtained good inpainting results. However, this method ignores the correlation between the missing areas of the image, so the inpainting results appear as a partial fault phenomenon. On the basis of this model, Zeng et al. [12] presented a model that used the image pyramid encode network and introduced the attention transfer model to achieve image inpainting. However, it lacks a deep understanding of the image. Meanwhile, Liu et al. [24] replaced ordinary convolution with partial convolution and combined an automatic mask update mechanism to inpaint irregular damaged images. There were unreasonable parts in its mask update mechanism. To realize the pluralistic image inpainting, the method proposed by Zheng et al. [25] combined the characteristics of GAN and VAE to achieve the generation of multiple inpainted images. However, it was only suitable for human-face images. Recently, Zili et al. [11] introduced a semantic residual aggregation mechanism for inpainting high-resolution images. They generated high-frequency residuals of the missing regions by weighting and aggregating the residuals of the context patches. Then, they used the attention model to inpaint images. The method reduced the computational cost and produced high-quality inpainting results. Deep learning-based inpainting methods can learn the deep semantic features of the image and generate contextually semantically consistent inpainting results. However, they tend to ignore the visual appeal of the images.

*2.2. Semantic Pyramid Network*

The image pyramid [26] is a collection of images at different scales as multi-scale representations of images. The semantic pyramid [27] is built on the basis of the image pyramid, and the model is constructed as a semantic pyramidal generation: the low-level information contains fine features (texture details, etc.), while the high-level information overlays high-level semantic information. In recent years, the semantic pyramid has been used for image inpainting. Zeng et al. [12] proposed a pyramid context encoder network, which filled the holes from deep to shallow levels by using the features learned from the pyramid encoder. Although this structure took less time than other structures, the acquired sensory field was larger, and it was difficult to obtain finer features.

*2.3. Contextual Attention Model*

The contextual attention model [23] was first proposed to fill the missing regions of an image by using an attention mechanism. First, it gathered the attention scores by using the regional affinity between missing regions or external patches and then fills the missing regions by replicating and aggregating patches from the weighted context according to the attention score. In another study [12], the holes were filled by using contextual attention models layer by layer on the pyramid network. The attention score was calculated only once and used on multiple conventional layers in [11]. To obtain finer features, our model uses the contextual attention model from deep to shallow levels for inpainting images.

## 3. Semantic Residual Pyramid Network

The Semantic Residual Pyramid Network (SRPNet) uses the principle of the generative adversarial network to inpaint the images. The SRPNet consists of three parts: a residual pyramid encoder, a multi-layer decoder, and a multi-scale discriminators. Where the residual pyramid encoder and the multi-layer decoder constitute a generator for producing the inpainted image, an overview of its network structure is shown in Table A1. The multi-scale discriminators identify the inpainted image to determine whether the image is the "truth," and an overview of its network structure is shown in Table A2. During the training process of the network, the generator and discriminator constantly interplay and eventually reach a balance.

We describe the specifics of the residual pyramid encoder in Section 3.1; then, we introduce the attention transfer model in Section 3.2; subsequently, the details of the multi-layer decoder are shown in Section 3.3; finally, we describe the multi-scale discriminators in Section 3.4. Figure 2 illustrates the network architecture of our proposed SRPNet.

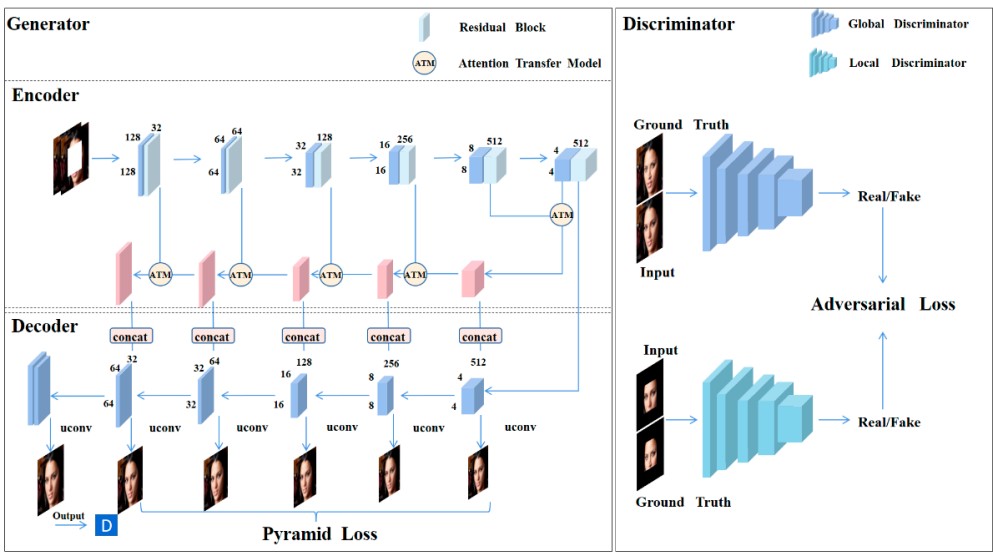

**Figure 2.** Model structure diagram of the Semantic Residual Pyramid Network (SRPNet). The SRPNet consists of a residual pyramid encoder, a multi-layer decoder, and multi-scale discriminators. The residual pyramid encoder and multi-layer decoder constitute the generator of the network, and the multi-scale discriminators are the discriminator of the network. The generator uses the masked image as input and inpaint the image layer by layer using the attention transfer model. Eventually, the inpainted image is the output. The multi-scale discriminators include a global discriminator and a local discriminator. The global discriminator takes the completed image as input, and the local discriminator takes the inpainted area of the output as input. The model is optimized by using pyramid loss, and the reconstruction loss is used to optimize the model and obtain finer images. (Best viewed with zoom-in).

### 3.1. Residual Pyramid Encoder

To obtain deeper semantic features, we generally attempt to deepen the network. However, this method may cause some problems, such as slow network convergence and increased errors. Thus, to a certain extent, this approach is unsuitable to increase the network depth. To solve these problems, our model uses instance normalization [16] to speed up the convergence of the model. Moreover, the residual blocks are added to each layer of the pyramid network to address the problem of reduced accuracy caused by the increase in the depth of the neural network. Therefore, we present the residual pyramid encoder built on the semantic pyramid structure [27]. It encodes the damaged images into compact latent features and decodes these features back to the image. The missing regions in the latent features are filled into a low-level feature layer with higher resolution and richer detail features. This design improves the effectiveness of the encoding further. In addition, the model fills the missing regions by reusing the attention transfer model (ATM) [12] from high-level semantic features to low-level features before decoding.

Suppose the depth of the encoder is n, and its feature maps from low to high are denoted as $f_1, \ldots, f_{n-2}, f_{n-1}, f_n$, respectively. The reconstructed feature maps of the ATM at each layer are represented as $F_{n-1}, F_{n-2}, \ldots, F_1$ from high to low:

$$
\begin{aligned}
F_{n-1} &= \psi\,(f_{n-1}, f_n), \\
F_{n-2} &= \psi\,(f_{n-2}, F_{n-1}), \\
&\ldots, \\
F_1 &= \psi\,(f_1, F_2) = \psi\,(f_1, \psi\,(f_2 \ldots, \psi\,(f_{n-1}, f_n))),
\end{aligned}
\tag{1}
$$

where $\psi$ represents the operation of the ATM. The missing pixels of the image are filled according to the semantic pyramid mechanism and the layer-by-layer attention transfer model. This design ensures the semantic consistency of the inpainted images. Its structure diagram is shown in Figure 3. Details regarding the specific operation of the ATM are introduced as follows.

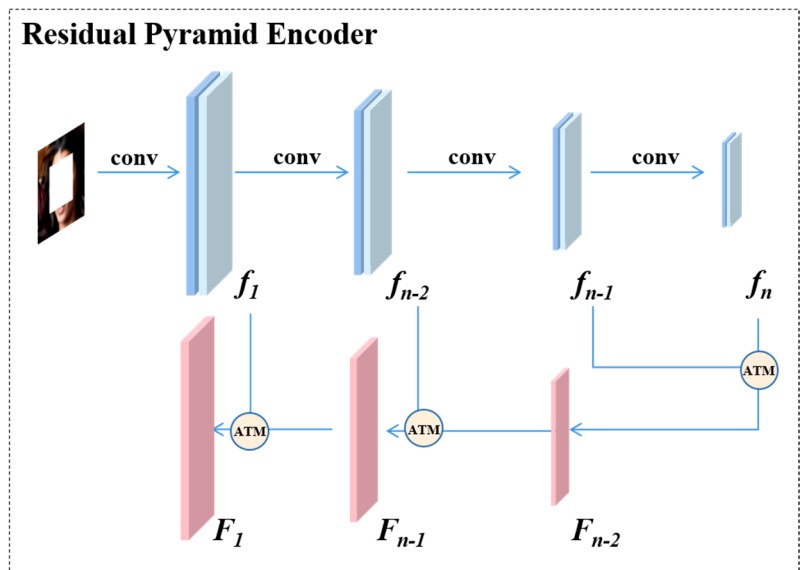

**Figure 3.** Residual Pyramid Encoder Model. First, the feature maps $(f_1, \ldots, f_{n-1}, f_{n-2}, f_n)$ of the original masked images are obtained by the residual pyramid encoder. Then, the missing pixels are filled by the Attention Transfer Model. Finally, we obtained the inpainted feature maps $(F_{n-1}, F_{n-2}, \ldots, F_1)$. (Best viewed in color).

### 3.2. Attention Transfer Model

First, the ATM learns its regional affinity from the high-level semantic feature graph ($f_n$). It extracts the patches for the missing region (p) from $f_n$ and calculates the cosine similarity ($Similarity_n$) between its internal and external patches:

$$Similarity_n = < \frac{p_o^n}{\|p_o^n\|}, \frac{p_i^n}{\|p_i^n\|} > \qquad (2)$$

where $p_o^n$ denotes the *o*-th patch extracted from outside the missing regions of $f_n$, and $p_i^n$ denotes the *i*-th patch extracted from inside the missing regions of $f_n$. Then, we used SoftMax on the similarity score to obtain the attention score ($Attention_n$) of each patch:

$$Attention_n = \frac{\exp(Similarity_n)}{\sum_{i=1}^N \exp(Similarity_n)} \qquad (3)$$

After obtaining the attention scores from the high-level semantic feature maps ($f_n$), we filled the missing regions of the adjacent low-level feature maps ($f_{n-1}$) through the context weighted by the attention score:

$$p_i^{n-1} = \sum_{i=1}^N Attention_n \bullet p_o^{n-1} \qquad (4)$$

where $p_i^{n-1}$ represents the *i*-th patch extracted from outside the missing regions of $f_{n-1}$, and $p_o^{n-1}$ represents the *o*-th patch in the missing regions of $f_{n-1}$ that need to be filled.

This operation was repeated to compute all patches and then filled $f_{n-1}$ afterward. Finally, we can obtain a completed feature map, $f_{n-1}$, which was obtained. The network structure of the ATM is shown in Figure 4. When applying the ATM to the semantic pyramid structure across layers, finer semantic features were obtained, and the context consistency of the filled feature map was ensured. Thus, the inpainting effect was improved further.

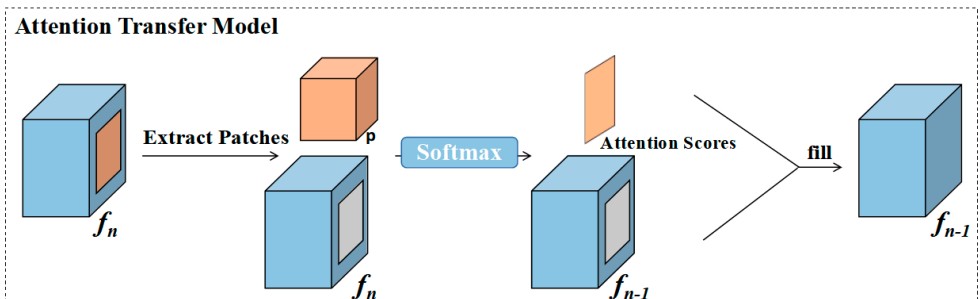

**Figure 4.** Attention Transfer Model. The patch (p) was extracted from the feature map ($f_n$) and calculated its attention scores by learning the region affinity. Subsequently, it used the attention scores to weigh the context for filling the feature map ($f_{n-1}$). Last, the inpainted feature map ($f_{n-1}$) was obtained. (Best viewed in color).

### 3.3. Multi-Layer Decoder

The multi-layer decoder first takes the semantic feature map, $f_n$, of the highest layer as the input of the decoder and decodes to obtain the feature map, $\eta_{n-1}$. Next, the inpainted feature map, $f_{n-1}$, from the ATM and the decoded feature map, $\eta_{n-1}$, are combined to obtain new feature maps, which are, represented as $\eta_{n-2}, \eta_{n-3} \ldots, \eta_1$, respectively. We sequentially took those feature maps as the input of the decoder and decoded them to acquire the predicted images of each layer, $X_1, X_2 \ldots, X_n$. Last, the pyramid loss [12] was used to optimize the image. The pyramid loss was used to gradually refine the final output by calculating the normalized L1 distance between the output at each scale and the original image. This method was used to improve the filling prediction of the missing regions at each scale. The network structure diagram of the multi-layer decoder is shown in Figure 2.

### 3.4. Multi-Scale Discriminators

Image inpainting is an unstable problem, so there are many different results for the missing regions. Therefore, we used the GAN [22] to select the image closest to the real image. The GAN contains at least one generator (G) and one discriminator (D). The role of the generator is to produce images based on the learned features, and the role of the discriminator is to judge whether the image produced by the generator is "real." The discriminator makes it difficult to distinguish the "fake" image produced by the generator from the real image by constant updating.

Our discriminator consists of a global discriminator and a local discriminator. The global discriminator takes the inpainted image as input to judge the consistency between the generated image and the ground truth. The local discriminator takes the inpainted missing regions as input for judging the semantic consistency of the local details. Compared with [10], the difference in our method is that the missing regions of the image become inpainted regions. The inpainted region can be either a central square or an irregularly shaped missing region. Our proposed method ensures not only the overall consistency of the image but also the consistency of the inpainted regions. The experiments prove that our method can generate excellent results both semantically and visually.

## 4. Experiments and Analysis

We present the experimental settings in Section 4.1; and the experimental results in Section 4.2, and we analyze the effectiveness of our model in Section 4.3.

### 4.1. Experimental Settings

We trained and tested four datasets with different characteristics: CELEBA-HQ [19], DTD [17], Facade [18], and Places2 [20]. Their characteristics are as follows, and the details are shown in Table 1. CELEBA-HQ [19] is high-quality face data from CELEBA [28]. DTD [17] is textured-image data containing 5640 images. It was classified into 47 categories according to human perception or texture diversity, with 120 images per category. Facade [18] is a collection of highly structured buildings from all over the world. Places2 [20] is a natural-scene dataset with 1,839,960 images, which can be divided into 365 different categories according to the different natural scenes. Additionally, we used the irregular mask dataset proposed by the Nvidia research team [24]. They divided the masks into six categories based on the size of the holes. Each category contains 1000 masks with and without boundary constraints, for a total of 12,000 masks.

**Table 1.** The details of the four different scene datasets.

| Datasets | Training | Testing | Total |
| --- | --- | --- | --- |
| DTD [17] | 4512 | 1128 | 5640 |
| Façade [18] | 506 | 100 | 606 |
| CELEBA-HQ [19] | 28,000 | 2000 | 30,000 |
| Places2 [20] | 1,829,960 | 10,000 | 1,839,960 |
| Mask [24] | − | − | 12,000 |

All of our experiments were trained and tested on $256 \times 256$ images. Our model ran on an NVIDIA GeForce RTX 3090 with a batch size of 64. We used a learning rate of $10^{-4}$ and a decay rate of 0.1. We used the Adam optimizer [29] with beta (0.5,0.999). Our code used TensorboardX [30] to observe when the model converges. The code was implemented in PyTorch [31].

We chose three classical methods in the field of image inpainting as the baseline for comparison: pluralistic image completion (PIC-NET) [25], GLCIC [9], and the pyramid-context encoder network (PEN-NET) [12]. Pluralistic image completion (PIC-NET) [25] is a method of inpainting images from diversified aspects, with the ability to generate multiple

visually realistic images. We utilized the code given by the author for training and testing. GLCIC [9] is a deep learning model that was the first to use a global discriminator and a local discriminator to inpaint images. It yields semantically consistent inpainting results. We used the official code for testing. The pyramid-context encoder network (PEN-NET) [12] is the first image pyramid encoder network to inpaint the semantic and visual aspects of images. It can produce semantically consistent and visually realistic inpainting results. We used the officially proposed pretraining model for testing.

*4.2. Experiments Results*

**Qualitative Comparisons.** We tested the images with $128 \times 128$ regular square masks and the different irregular masks. Compared with the most classical models, our model exhibits semantic consistency and excellent visual effects. As shown in the figure, PIC-NET [25] can generate clear and complete images for the partial datasets but lacks semantic consistency compared to real images. Additionally, this model is not applicable for all scene images. GLCIC [10] generates blurring images for large-area mask occlusion and produces an image that is obviously inconsistent with the ground truth for inpainting the irregular regions. Moreover, the images generated by PEN-NET [12] contain blurred visual effects and partial distortion. Instead, as can be seen from the figure, our method can generate visually similar but also semantically consistent results for the inpainting of the square and irregular missing regions. The comparison of our experimental results for the regular mask and the irregular mask are shown in Figure 5. (We give the rest of the experimental results in Appendix A Figure A1).

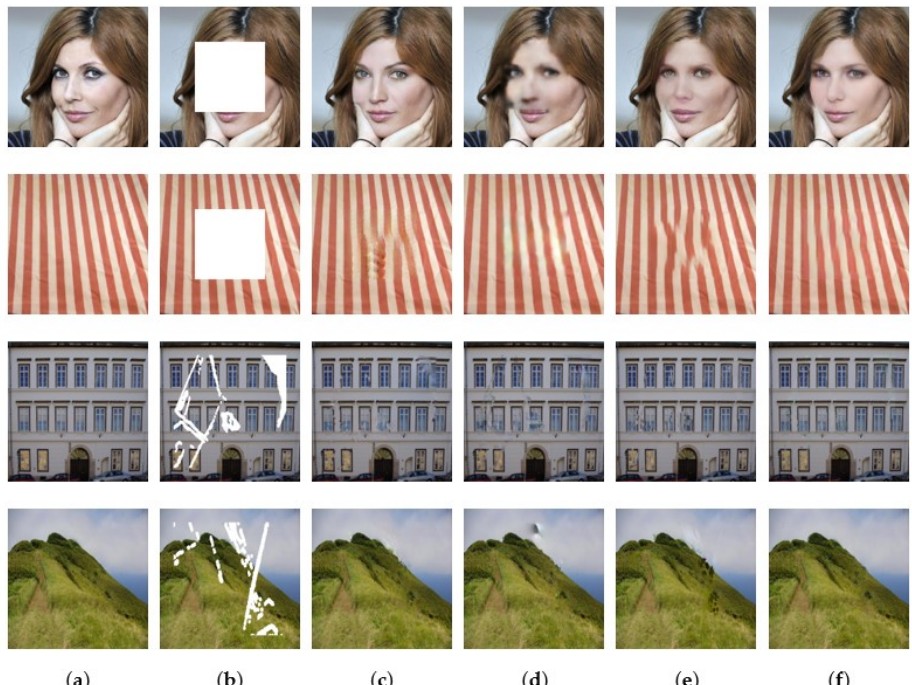

|     |     |     |     |     |     |
| --- | --- | --- | --- | --- | --- |
| (a) | (b) | (c) | (d) | (e) | (f) |

**Figure 5.** Qualitative comparisons for image inpainting with square mask and irregular mask on four different datasets. From left to right: the original image, the input image, the results of the baseline model, and our results. (Best viewed with zoom-in). (**a**) GT; (**b**) Input; (**c**) PIC-NET; (**d**) GLCIC; (**e**) PEN-NET; (**f**) Ours.

**Quantitative Comparisons.** We used the L1 loss, the peak-signal-to-noise ratio (PSNR) [32], and the multi-scale structural-similarity-index measure (MS-SSIM) [33] as evaluation metrics for quantitative evaluation. The L1 loss is the sum of the absolute differences between the pixel values of the predicted image and the actual pixel values of the ground truth. It reflects the actual error between the predicted image and the real image. PSNR [32] is the most commonly and widely used image objective-evaluation indicator that measures

the difference between the pixel values of two images. MS-SSIM [33] is used to measure the similarity of images at different resolutions. In addition to the objective-evaluation data, we also performed a quantitative comparison of four different datasets: DTD [17], Facade [18], CELEBA-HQ [19], and Places2 [20]. All images were trained images with a 128 × 128 regular square mask and an irregular mask [24]. The images we tested were output directly from our trained models without any processing of the already inpainted images. The inpainting-effect data of the model on the regular square mask are shown in Table 2, and the inpainting data of the irregular mask are shown in Table 3. Meanwhile, we conducted experiments on different models for four different datasets. The purpose of the experiments was to test the ability of the model to inpaint irregular masks of different sizes, for which the part of the experimental data are shown in Table 4. (We give all the experimental data in Appendix A Tables A3–A5.) As seen from the data in Table 2, our method numerically outperformed the comparison methods on all four datasets. Among them, the data from the MS-SSIM [33] improved more than other evaluation indicators, which verifies the effectiveness of the added multi-scale components. Our experimental data on Façade [18] and Places2 [20] datasets were more prominent. It indicated that our method is more applicable for inpainting of the natural-scene images. As can be seen from the data in Table 3, although the inpainting effects of our method on the DTD [17] dataset are still insufficient, its experimental data on other datasets significantly outperform the comparison methods. It can be concluded that our model shows better performance in both regular masks and irregular masks. Observing the data in Table 4, it can be seen that our method has better inpainting effects on the larger missing areas. This also verifies that our model is more suitable for inpainting large holes.

**Table 2.** The inpainting-effect data of the central square mask. It includes a quantitative comparison between L1, PSNR [32], and MS-SSIM [33] on four different datasets. ↑ higher is better; ↓ lower is better.

| Datasets | Methods | $L_1$ (↓) | PSNR (↑) | MS-SSIM (↑) |
|---|---|---|---|---|
| DTD | PIC-NET | 3.72% | 22.56 | 72.69% |
| | GLCIC | 3.49% | 23.48 | 73.78% |
| | PEN-NET | 3.44% | 23.73 | 75.13% |
| | Ours | **3.25%** | **24.17** | **76.62%** |
| Facade | PIC-NET | 4.04% | 20.92 | 74.40% |
| | GLCIC | 3.63% | 21.87 | 77.87% |
| | PEN-NET | 3.69% | 21.87 | 77.59% |
| | Ours | **3.02%** | **23.22** | **82.49%** |
| CELEBA-HQ | PIC-NET | 2.69% | 24.29 | 87.26% |
| | GLCIC | 2.63% | 24.96 | 88.94% |
| | PEN-NET | 2.40% | 25.47 | 89.03% |
| | Ours | **2.18%** | **26.31** | **90.76%** |
| Places2 | PIC-NET | 3.10% | 22.64 | 76.78% |
| | GLCIC | 2.76% | 22.96 | 73.39% |
| | PEN-NET | 2.75% | 23.80 | 78.68% |
| | Ours | **2.58%** | **24.37** | **80.52%** |

**Table 3.** The inpainting effect data of the irregular mask. It includes a quantitative comparison between L1, PSNR [32], and MS-SSIM [33] on four different datasets. ↑ higher is better;↓ lower is better.

| Datasets | Methods | $L_1$ (↓) | PSNR (↑) | MS-SSIM (↑) |
|---|---|---|---|---|
| DTD | PIC-NET | 0.82% | 31.14 | 95.83% |
| | GLCIC | 1.92% | 29.66 | 93.51% |
| | PEN-NET | **0.62%** | **33.01** | 96.28% |
| | Ours | 0.67% | 31.75 | **96.35%** |

**Table 3.** *Cont.*

| Datasets | Methods | $L_1$ (↓) | PSNR (↑) | MS-SSIM (↑) |
|---|---|---|---|---|
| Facade | PIC-NET | 0.86% | 29.28 | 96.28% |
| | GLCIC | 0.98% | 28.04 | 94.87% |
| | PEN-NET | 0.58% | 31.98 | 97.60% |
| | Ours | **0.53%** | **32.19** | **97.85%** |
| CELEBA-HQ | PIC-NET | 0.46% | 35.68 | 98.93% |
| | GLCIC | 1.35% | 28.65 | 93.37% |
| | PEN-NET | 0.84% | 32.06 | 96.94% |
| | Ours | **0.45%** | **35.73** | **98.99%** |
| Places2 | PIC-NET | 2.17% | 27.47 | 87.00% |
| | GLCIC | 1.42% | 31.72 | 94.79% |
| | PEN-NET | 0.83% | 31.11 | 94.83% |
| | Ours | **0.67%** | **32.32** | **96.08%** |

**Table 4.** Different degrees of masks compared to the quantitative comparison of L1, PSNR [32], and MS-SSIM [33] in CELEBA-HQ [19]. ↑ higher is better; ↓ lower is better.

| Mask | Methods | $L_1$ (↓) | PSNR (↑) | MS-SSIM (↑) |
|---|---|---|---|---|
| [0.01,0.1] | PIC-NET | 0.41% | 35.67 | 98.93% |
| | GLCIC | 1.24% | 32.09 | 96.94% |
| | PEN-NET | 0.48% | 33.56 | 98.48% |
| | Ours | **0.36%** | **35.73** | **98.99%** |
| (0.1,0.2] | PIC-NET | 1.06% | 30.13 | 96.85% |
| | GLCIC | 3.59% | 24.70 | 89.55% |
| | PEN-NET | 1.34% | 27.85 | 95.40% |
| | Ours | **0.96%** | **30.42** | 97.13% |
| (0.2,0.3] | PIC-NET | 1.92% | 26.94 | 93.84% |
| | GLCIC | 6.45% | 20.53 | 78.02% |
| | PEN-NET | 2.46% | 24.73 | 91.13% |
| | Ours | **1.75%** | **27.35** | **94.44%** |
| (0.3,0.4] | PIC-NET | 2.94% | 24.52 | 89.79% |
| | GLCIC | 9.46% | 17.84 | 65.20% |
| | PEN-NET | 3.76% | 22.57 | 86.10% |
| | Ours | **2.67%** | **25.11** | **91.13%** |
| (0.4,0.5] | PIC-NET | 4.18% | 22.44 | 84.53% |
| | GLCIC | 12.19% | 16.12 | 53.44% |
| | PEN-NET | 5.42% | 20.66 | 79.29% |
| | Ours | **3.77%** | **23.23** | **86.94%** |
| (0.5,0.6] | PIC-NET | 6.73% | 19.41 | 73.08% |
| | GLCIC | 15.38% | 14.68 | 41.36% |
| | PEN-NET | 7.76% | 18.72 | 69.40% |
| | Ours | **5.69%** | **20.79** | **78.67%** |

*4.3. Ablation Study*

We used the ablation experiments to determine the validity of our proposed network. To verify the effectiveness of the residual blocks and the multi-scale discriminators, we added the residual blocks and the multi-scale discriminators to the baseline model in turn. We trained and tested the effects of inpainting the central square missing region on the Facade [18] and CELEBA-HQ [19]. The test data are shown in Table 5, where we can observe that the inpainting ability of the models gradually improved with the increase in components. At the same time, we show the test results under different models in Figure 6. As seen from the figure, the baseline output has different degrees of blurring. After adding the discriminator, some of the blurred areas were eliminated. Finally, when the residual blocks were added to the model, the output image had a good visual effect.

Therefore, we can prove the effectiveness of the aforementioned addition of the components for image inpainting.

**Table 5.** Ablation comparison of the discriminator and residual blocks over Facade [18] and CELEBA-HQ [19]. ↑ higher is better; ↓ lower is better.

| Datasets | Methods | $L_1$ (↓) | PSNR (↑) | MS-SSIM (↑) |
|---|---|---|---|---|
| Facade | Baseline | 3.69% | 21.87 | 77.59% |
| | +Discriminator | **3.02%** | 22.47 | 80.11% |
| | +Residual blocks (ours) | 3.36% | **23.22** | **82.49%** |
| CELEBA-HQ | Baseline | 2.40% | 25.47 | 89.03% |
| | +Discriminator | 2.39% | 25.43 | 88.70% |
| | +Residual blocks (ours) | **2.18%** | **26.27** | **90.64%** |

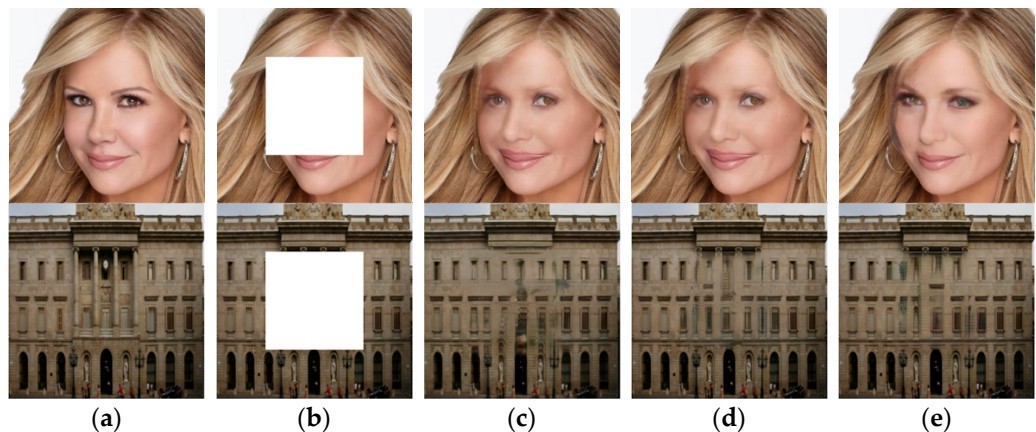

| (**a**) | (**b**) | (**c**) | (**d**) | (**e**) |

**Figure 6.** Comparison of different training-model outputs. From left to right: the original image, the input image, the results of the baseline model, the results of adding the discriminator model, and our results. (Best viewed with zoom-in). (**a**) GT; (**b**) Input; (**c**) Baseline; (**d**) +Discriminator; (**e**) +Residual blocks.

## 5. Conclusions

In this article, we proposed a Residual Semantic Pyramid Network (SRPNet) based on the image inpainting model to inpaint missing regions of images. Our approach was based on the GAN. The generator generated an image, and then the discriminator judged whether the generated image was "real." In order to acquire more semantic features, we designed a residual pyramid encoder and put it into the generator network. We also introduced a new discriminator for improving the ability of the model to determine the local semantics of the image. The experiments showed that our approach could generate images with consistent semantics and realistic visual effects on different datasets. In the future, we will focus on the high-resolution image inpainting and improve the visual quality of the inpainting results.

**Author Contributions:** Conceptualization, H.L. and Y.Z.; methodology, H.L.; software, H.L.; validation, H.L. and Y.Z.; formal analysis, H.L.; investigation, H.L.; resources, Y.Z.; data curation, H.L.; writing—original draft preparation, H.L. and Y.Z.; writing—review and editing, H.L. and Y.Z.; visualization, H.L.; supervision, Y.Z.; project administration, Y.Z.; funding acquisition, Y.Z. All authors have read and agreed to the published version of the manuscript.

**Funding:** This research was supported in part by the National Natural Science Foundation of China under Grants 61972206 and 62011540407; in part by the Natural Science Foundation of Jiangsu Province under Grant BK20211539; in part by the 15th Six Talent Peaks Project in Jiangsu Province under Grant RJFW-015; in part by the Qing Lan Project, in part by the PAPD fund.

**Institutional Review Board Statement:** Not applicable.

**Informed Consent Statement:** Not applicable.

**Data Availability Statement:** The data presented in this study are available on request from the corresponding author. Our code can be found at https://github.com/luobo348/SRPNet.git (accessed on 4 January 2022).

**Acknowledgments:** I want to thank my teacher for supporting and encouraging my research project.

**Conflicts of Interest:** The authors declare no conflict of interest.

## Appendix A

**Table A1.** Generator network structure summary of SRPNet. The generator consists of Residual Pyramid Encoder and Multi-layer Decoder.

| Type | Structure |
| --- | --- |
| Input | $4 \times 256 \times 256$ (Image + Mask) |
| Conv | input: 4, kernel: 3, stride: 2, padding: 1, LReLU, output: 32 |
| Residual_blocks | input: 32, kernel: 3, stride: 2, padding: 0, ReLU, output: 32 |
| Conv | input: 32, kernel: 3, stride: 2, padding: 1, LReLU, output: 64 |
| Residual_blocks | input: 64, kernel: 3, stride: 2, padding: 0, ReLU, output: 64 |
| Conv | input: 64, kernel: 3, stride: 2, padding: 1, LReLU, output: 128 |
| Residual_blocks | input: 128, kernel: 3, stride: 2, padding: 0, ReLU, output: 128 |
| Conv | input: 128, kernel: 3, stride: 2, padding: 1, LReLU, output: 256 |
| Residual_blocks | input: 256, kernel: 3, stride: 2, padding: 0, ReLU, output: 256 |
| Conv | input: 256, kernel: 3, stride: 2, padding: 1, LReLU, output: 512 |
| Residual_blocks | input: 512, kernel: 3, stride: 2, padding: 0, ReLU, output: 512 |
| Conv | input: 512, kernel: 3, stride: 2, padding: 1, LReLU, output: 512 |
| Residual_blocks | input: 512, kernel: 3, stride: 2, padding: 0, ReLU, output: 512 |
| ATMConv | input: 512, kernel: 1, stride: 1, output: 512 |
| ATMConv | input: 256, kernel: 1, stride: 1, output: 256 |
| ATMConv | input: 128, kernel: 1, stride: 1, output: 128 |
| ATMConv | input: 64, kernel: 1, stride: 1, output: 64 |
| ATMConv | input: 32, kernel: 1, stride: 1, output: 32 |
| DeConv | input: 512, kernel: 3, stride: 1, padding: 1, ReLU, output: 512 |
| DeConv | input: 1024, kernel: 3, stride: 1, padding: 1, ReLU, output: 256 |
| DeConv | input: 512, kernel: 3, stride: 1, padding: 1, ReLU, output: 128 |
| DeConv | input: 256, kernel: 3, stride: 1, padding: 1, ReLU, output: 64 |
| DeConv | input: 128, kernel: 3, stride: 1, padding: 1, ReLU, output: 32 |
| Output1 | input: 1024, kernel: 1, stride: 1, padding: 0, Tanh, output: 3 |
| Output2 | input: 512, kernel: 1, stride: 1, padding: 0, Tanh, output: 3 |
| Output3 | input: 256, kernel: 1, stride: 1, padding: 0, Tanh, output: 3 |
| Output4 | input: 128, kernel: 1, stride: 1, padding: 0, Tanh, output: 3 |
| Output5 | input: 64, kernel: 1, stride: 1, padding: 0, Tanh, output: 3 |
| Output6 | input: 64, kernel: 3, stride: 1, padding: 1, ReLU, output: 32input: 32, kernel: 3, stride: 1, padding: 1, Tanh, output: 3 |

**Table A2.** Discriminator network structure summary of SRPNet. Our model uses the same network structure of the global discriminator and the local discriminator.

| Type | Structure |
| --- | --- |
| Conv | input: 3, kernel: 5, stride: 2, padding: 1, LReLU, output: 64 |
| Conv | input: 64, kernel: 5, stride: 2, padding: 1, LReLU, output: 128 |
| Conv | input: 128, kernel: 5, stride: 2, padding: 1, LReLU, output: 256 |
| Conv | input: 256, kernel: 5, stride: 2, padding: 1, LReLU, output: 512 |
| Conv | input: 512, kernel: 5, stride: 2, padding: 1, LReLU, output: 1 |

**Table A3.** Different degrees of masks compared to the quantitative comparison of L1, PSNR [32], and MS-SSIM [33] in DTD [17]. ↑ higher is better; ↓ lower is better.

| Mask | Methods | $L_1$ (↓) | PSNR (↑) | MS-SSIM (↑) |
|---|---|---|---|---|
| [0.01,0.1] | PIC-NET | 0.83% | 31.14 | 95.83% |
| | GLCIC | 1.01% | 32.67 | 96.52% |
| | PEN-NET | 0.67% | 32.37 | 96.57% |
| | Ours | **0.66%** | **32.75** | **96.55%** |
| (0.1,0.2] | PIC-NET | 2.10% | 26.07 | 90.88% |
| | GLCIC | 2.87% | 26.36 | 89.82% |
| | PEN-NET | 1.79% | 26.72 | 91.91% |
| | Ours | **1.76%** | **26.78** | **92.01%** |
| (0.2,0.3] | PIC-NET | 3.60% | 23.49 | 83.44% |
| | GLCIC | 5.19% | 22.60 | 78.60% |
| | PEN-NET | 3.27% | 23.68 | 84.29% |
| | Ours | **3.13%** | **24.05** | **85.32%** |
| (0.3,0.4] | PIC-NET | 5.22% | 21.58 | 75.08% |
| | GLCIC | 7.41% | 20.08 | 66.32% |
| | PEN-NET | 4.90% | 21.65 | 75.42% |
| | Ours | **4.59%** | **22.23** | **77.89%** |
| (0.4,0.5] | PIC-NET | 7.11% | 19.94 | 65.36% |
| | GLCIC | 9.51% | 18.39 | 54.08% |
| | PEN-NET | 6.61% | 20.17 | 65.90% |
| | Ours | **6.24%** | **20.70** | **69.23%** |
| (0.5,0.6] | PIC-NET | 9.90% | 17.85 | 51.65% |
| | GLCIC | 11.51% | 16.95 | 40.43% |
| | PEN-NET | 8.61% | 18.81 | 54.91% |
| | Ours | **8.18%** | **19.26** | **57.48%** |

**Table A4.** Different degrees of masks compared to the quantitative comparison of L1, PSNR [32], and MS-SSIM [33] in Facade [18]. ↑ higher is better; ↓ lower is better.

| Mask | Methods | $L_1$ (↓) | PSNR (↑) | MS-SSIM (↑) |
|---|---|---|---|---|
| [0.01,0.1] | PIC-NET | 0.86% | 29.28 | 96.28% |
| | GLCIC | 0.98% | 31.48 | **97.92%** |
| | PEN-NET | 0.58% | 31.98 | 97.60% |
| | Ours | **0.54%** | **32.20** | 97.85% |
| (0.1,0.2] | PIC-NET | 2.68% | 22.89 | 90.81% |
| | GLCIC | 2.71% | 25.38 | 84.75% |
| | PEN-NET | 1.75% | 24.94 | 93.43% |
| | Ours | **1.45%** | **26.34** | **95.19%** |
| (0.2,0.3] | PIC-NET | 4.81% | 20.04 | 82.11% |
| | GLCIC | 4.76% | 19.98 | 76.06% |
| | PEN-NET | 3.40% | 21.67 | 86.66% |
| | Ours | **2.71%** | **23.41** | **89.33%** |
| (0.3,0.4] | PIC-NET | 6.85% | 18.48 | 72.83% |
| | GLCIC | 6.67% | 19.98 | 76.06% |
| | PEN-NET | 5.10% | 19.84 | 79.35% |
| | Ours | **4.33%** | **20.92** | **82.36%** |
| (0.4,0.5] | PIC-NET | 9.11% | 17.02 | 63.72% |
| | GLCIC | 9.06% | 18.21 | 66.51% |
| | PEN-NET | 6.89% | 18.45 | 70.21% |
| | Ours | **6.16%** | **19.19** | **74.51%** |

**Table A4.** *Cont.*

| Mask | Methods | L$_1$ (↓) | PSNR (↑) | MS-SSIM (↑) |
|---|---|---|---|---|
| (0.5,0.6] | PIC-NET | 11.88% | 15.53 | 50.26% |
| | GLCIC | 11.16% | 16.65 | 53.59% |
| | PEN-NET | 9.27% | 16.95 | 59.52% |
| | Ours | **8.20%** | **17.67** | **64.01%** |

**Table A5.** Different degrees of masks compared to the quantitative comparison of L1, PSNR [32], and MS-SSIM [33] in Places2 [20]. ↑ higher is better; ↓ lower is better.

| Mask | Methods | L$_1$ (↓) | PSNR (↑) | MS-SSIM (↑) |
|---|---|---|---|---|
| [0.01,0.1] | PIC-NET | 0.45% | **34.71** | 97.53% |
| | GLCIC | 0.74% | 32.11 | 95.64% |
| | PEN-NET | 0.49% | 33.61 | 97.06% |
| | Ours | **0.41%** | 34.35 | **97.54%** |
| (0.1,0.2] | PIC-NET | 1.16% | 29.37 | 93.48% |
| | GLCIC | 2.10% | 28.29 | 91.87% |
| | PEN-NET | 1.35% | 27.49 | 91.47% |
| | Ours | **1.06%** | **29.38** | **93.84%** |
| (0.2,0.3] | PIC-NET | 2.02% | 26.40 | 88.11% |
| | GLCIC | 3.77% | 24.45 | 83.00% |
| | PEN-NET | 2.46% | 24.36 | 84.25% |
| | Ours | **1.89%** | **26.47** | **88.73%** |
| (0.3,0.4] | PIC-NET | 2.99% | 24.28 | 81.80% |
| | GLCIC | 5.52% | 21.80 | 72.63% |
| | PEN-NET | 3.76% | 22.28 | 76.27% |
| | Ours | **2.82%** | **24.47** | **82.64%** |
| (0.4,0.5] | PIC-NET | 4.16% | 22.41 | 73.96% |
| | GLCIC | 7.17% | 19.90 | 62.01% |
| | PEN-NET | 5.42% | 20.55 | 66.94% |
| | Ours | **3.98%** | **22.56** | **74.94%** |
| (0.5,0.6] | PIC-NET | 6.45% | 19.55 | 58.67% |
| | GLCIC | 9.03% | 18.01 | 49.70% |
| | PEN-NET | 7.76% | 19.13 | 56.79% |
| | Ours | **5.73%** | **20.44** | **63.27%** |

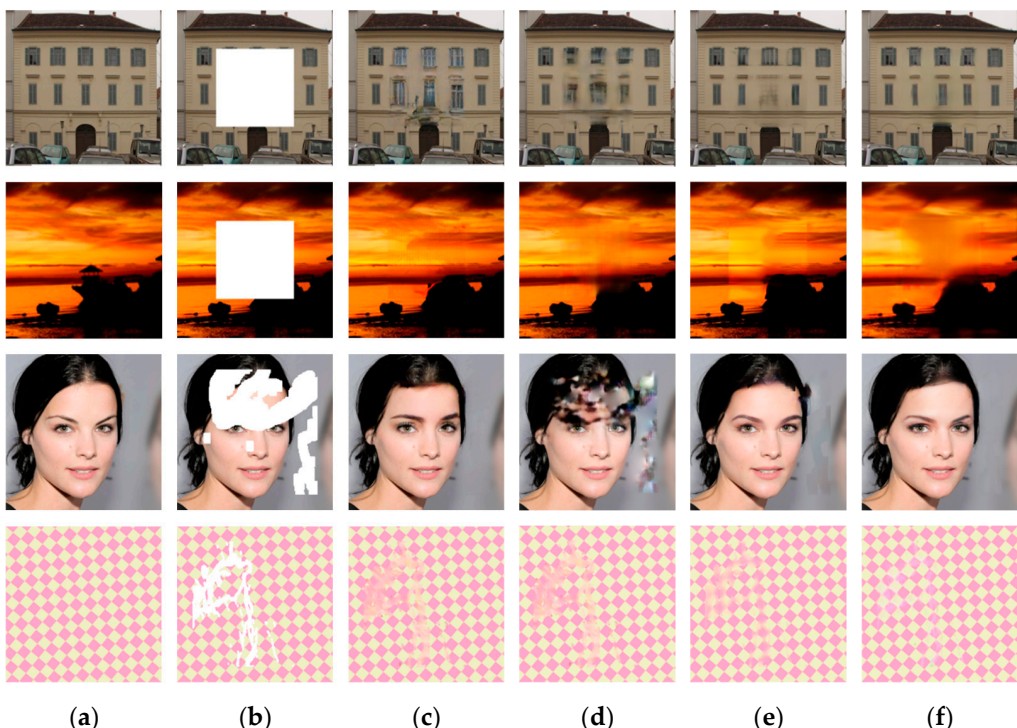

|       |       |       |       |       |       |
| :---: | :---: | :---: | :---: | :---: | :---: |
| (**a**) | (**b**) | (**c**) | (**d**) | (**e**) | (**f**) |

**Figure A1.** Qualitative comparisons for image inpainting with irregular masks on four different datasets. From left to right: the original image, the input image, the results of the baseline model, and our results. (Best viewed with zoom-in). (**a**) GT; (**b**) Input; (**c**) PIC-NET; (**d**) GLCIC; (**e**) PEN-NET; (**f**) Ours.

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
