# Peer review of "Semantic Residual Pyramid Network for Image Inpainting"

_information, doi:10.3390/info13020071_

Round 1

Reviewer 1 Report

This paper proposes a Semantic Residual Pyramid Network to solve the image inpainting problem. The authors propose the use of residual blocks and multi-scale discriminator-based generating adversarial networks for more consistent and realistic predictions. The authors have shown better quantitative results on four standard datasets compared to the recent methods.

Strengths:

  • I liked the idea of the Attention Transfer Model which ensures the consistency of the context in the filled feature map region. This idea might be of interest to the research community.
  • Quantitative results consistently outperform the previous methods.

Weaknesses:

  • The writing of this paper can be improved. The presentation in the paper especially the quality of the results is not clear to claim the effectiveness of the paper.
  • Please correct the author name to Iizuka et al. (line 107). 
  • Please add other recent works on image inpainting [12,15, 25, 27, 31]  with explanations in Section 2.1.
  • The images are of low resolution and hard to decipher. e.g. Fig 2 is blurry and difficult to read. Please provide better resolution figures.
  • The authors mention that they have trained and tested the network on 256*256 images (line 256). However, the qualitative results shown in this manuscript are of low resolution. Please provide the image results with 256*256 resolution for better understanding. 
  • Why do the authors mention in the conclusion that the “method was generally applicable to low-resolution images” (line 377)?

Author Response

For your suggestion, I have answered it in the documentation. Please see the attachment.

Reviewer 2 Report

The paper is well-written and presents promising results. The method's description is clear, the paper's novelty is sufficient, and satisfactory theoretical results have also been provided. The simulation results are quite convincing. At the same time, there are some remarks to be corrected.

The manuscript should be spell-checked and grammar-checked.

Lines 51-57: the text is identical to the abstract. It must be reformulated.

What do the authors mean by saying "the semantic consistency"?

Section 4.1: the first paragraph must be substituted by a table. It is a better way of presenting information for this case.

Section 4.1, the last paragraph: the authors mention "the classic methods of image inpainting," but they do not specify these methods precisely. It looks reasonable to enumerate these methods in the first sentence and then to describe each procedure in the same abstract.

Tables 1-3: the in-depth analysis is required here, including some suggestions and assumptions why a specific solution has demonstrated a particular result.

Would the authors mind adding a link to the source code of the developed network?

Author Response

(The authors gave the same response as above.)
